# Composing Complex Skills by Learning Transition Policies

**Youngwoon Lee**\*, **Shao-Hua Sun**\*, **Sriram Somasundaram, Edward S. Hu, Joseph J. Lim**
University of Southern California
{lee504,shaohuas,sriramso,hues,limjj}@usc.edu

## Abstract

Humans acquire complex skills by exploiting previously learned skills and making transitions between them. To empower machines with this ability, we propose a method that can learn transition policies which effectively connect primitive skills to perform sequential tasks without handcrafted rewards. To efficiently train our transition policies, we introduce proximity predictors which induce rewards gauging proximity to suitable initial states for the next skill. The proposed method is evaluated on a set of complex continuous control tasks in bipedal locomotion and robotic arm manipulation which traditional policy gradient methods struggle at. We demonstrate that transition policies enable us to effectively compose complex skills with existing primitive skills. The proposed induced rewards computed using the proximity predictor further improve training efficiency by providing more dense information than the sparse rewards from the environments. We make our environments, primitive skills, and code public for further research at https://youngwoon.github.io/transition.

## 1 Introduction

While humans are capable of learning complex tasks by reusing previously learned skills, composing and mastering complex skills are not as trivial as sequentially executing those acquired skills. Instead, it requires a smooth transition between skills since the final pose of one skill may not be appropriate to initiate the following one. For example, scoring in basketball with a quick shot after receiving a ball can be decomposed into catching and shooting. However, it is still difficult for beginners who have learned to catch passes and statically shoot. To master this skill, players must practice adjusting their footwork and body into a comfortable shooting pose after catching a pass.

Can machines similarly learn new and complex tasks by reusing acquired skills and learning transitions between them? Learning to perform composite and long-term tasks from scratch requires extensive exploration and sophisticated reward design, which can introduce undesired behaviors (Riedmiller et al., 2018). Thus, instead of employing intricate reward functions and learning from scratch, modular methods sequentially execute acquired skills with a rule-based meta-policy, enabling machines to solve complicated tasks (Pastor et al., 2009; Mülling et al., 2013; Andreas et al., 2017). These modular approaches assume that a task can be clearly decomposed into several subtasks which are smoothly connected to each other. In other words, an ending state of one subtask falls within the set of starting states, *initiation set*, of the next subtask (Sutton et al., 1999). However, this assumption does not hold in many continuous control problems where a given skill may be executed in starting states not considered during training or designing and thus, fail to achieve its goal.

To bridge the gap between skills, we propose a *transition policy* which learns to smoothly navigate from an ending state of a skill to suitable initial states of the following skill, as illustrated in Figure 1. However, learning a transition policy between skills without reward shaping is difficult as the only available learning signal is the sparse reward for the successful execution of the next skill. Sparse success/failure reward is challenging to learn from due to the temporal credit assignment problem (Sutton, 1984) and the lack of information from failing trajectories. To alleviate these problems, we propose a *proximity predictor* which outputs the proximity to the initiation set of the next skill and acts as a dense reward function for the transition policy.

---

\*Equal contribution

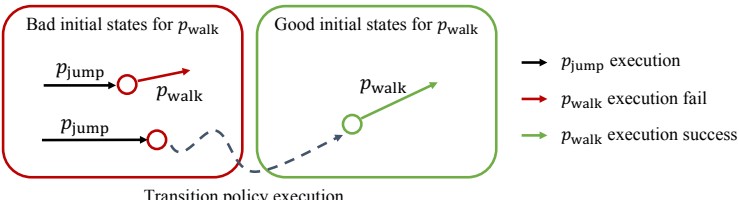

Figure 1: Concept of a transition policy. Composing complex skills using primitive skills requires smooth transition between primitive skills since a following primitive skill might not be robust to ending states of the previous one. In this example, the ending states (red circles) of the primitive policy $p_{jump}$ are not good initial states to execute the following policy $p_{walk}$. Therefore, executing $p_{walk}$ from these states will fail (red arrow). To smoothly connect the two primitive policies, we propose a transition policy which navigates an agent to suitable initial states for $p_{walk}$ (dashed arrow), leading to a successful execution of $p_{walk}$ (green arrow).

The main contributions of this paper include (1) the concept of learning transition policies to smoothly connect primitive skills; (2) a novel modular framework with transition policies that is able to compose complex skills by reusing existing skills; and (3) a joint training algorithm with the proximity predictor specifically designed for efficiently training transition policies. This framework is suited for learning complex skills that require sequential execution of acquired primitive skills, which are common for humans yet relatively unexplored in robot learning. Our experiments on simulated environments demonstrate that employing transition policies solves complex continuous control tasks which traditional policy gradient methods struggle at.

## 2 RELATED WORK

Learning continuous control of diverse behaviors in locomotion (Merel et al., 2017; Heess et al., 2017; Peng et al., 2017) and robotic manipulation (Ghosh et al., 2018) is an active research area in reinforcement learning (RL). While some complex tasks can be solved through extensive reward engineering (Ng et al., 1999), undesired behaviors often emerge (Riedmiller et al., 2018) when tasks require several different primitive skills. Moreover, training complex skills from scratch is not computationally practical.

Real-world tasks often require diverse behaviors and longer temporal dependencies. In hierarchical reinforcement learning, the option framework (Sutton et al., 1999) learns meta actions (options), a series of primitive actions over a period of time. Typically, a hierarchical reinforcement learning framework consists of two components: a high-level meta-controller and low-level controllers. A meta-controller determines the order of subtasks to achieve the final goal and chooses corresponding low-level controllers that generate a sequence of primitive actions. Unsupervised approaches to discover meta actions have been proposed (Schmidhuber, 1990; Daniel et al., 2016; Bacon et al., 2017; Vezhnevets et al., 2017; Dilokthanakul et al., 2017; Levy et al., 2017; Frans et al., 2018; Co-Reyes et al., 2018; Mao et al., 2018). However, to deal with more complex tasks, additional supervision signals (Andreas et al., 2017; Merel et al., 2017; Shu et al., 2018) or pre-defined low-level controllers (Kulkarni et al., 2016; Oh et al., 2017) are required.

To exploit pre-trained modules as low-level controllers, neural module networks (Andreas et al., 2016) have been proposed, which construct a new network dedicated to a given query using a collection of reusable modules. In the RL domain, a meta-controller is trained to follow instructions (Oh et al., 2017) and demonstrations (Xu et al., 2017), and support multi-level hierarchies (Gudimella et al., 2017). In the robotics domain, Pastor et al. (2009); Kober et al. (2010); Mülling et al. (2013) have proposed a modular approach that learns table tennis by selecting appropriate low-level controllers. On the other hand, Andreas et al. (2017); Frans et al. (2018) learn abstract skills while experiencing a distribution of tasks and then solve a new task with the learned primitive skills. However, these modular approaches result in undefined behavior when two skills are not smoothly connected. Our proposed framework aims to bridge this gap by training transition policies in a model-free manner to navigate the agent from unseen states for following skills to suitable initial states.

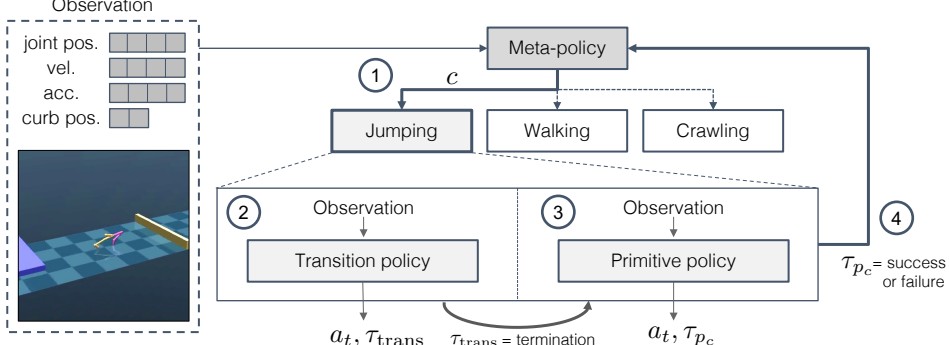

Figure 2: Our modular network augmented with transition policies. To perform a complex task, our model repeats the following steps: (1) The meta-policy chooses a primitive policy of index $c$; (2) The corresponding transition policy helps initiate the chosen primitive policy; (3) The primitive policy executes the skill; and (4) A success or failure signal for the primitive skill is produced.

Deep RL techniques for continuous control demand dense reward signals; otherwise, they suffer from long training time. Instead of manual reward shaping for denser reward, adversarial reinforcement learning (Ho & Ermon, 2016; Merel et al., 2017; Wang et al., 2017; Bahdanau et al., 2019) employs a discriminator which learns to judge the state or the policy, and the policy takes as rewards the output of the discriminator. While those methods assume ground truth trajectories or goal states are given, our method collects both success and failure trajectories online to train proximity predictors which provide rewards for transition policies.

## 3 APPROACH

In this paper, we address the problem of solving a complex task that requires sequential composition of primitive skills given only *sparse and binary rewards* (i.e. subtask completion reward). The sequential execution of primitive skills fails when two consecutive skills are not smoothly connected. We propose a modular framework with *transition policies* that learn to make transition between one policy to the subsequent policy, and therefore, can exploit the given primitive skills to compose complex skills. To accelerate training of transition policies, additional networks, *proximity predictors*, are jointly trained to provide *proximity rewards* as intermediate feedback to transition policies. In Section 3.2, we describe our framework in details. Next, in Section 3.3, we elaborate how transition policies are efficiently trained with induced proximity reward.

### 3.1 PRELIMINARIES

We formulate our problem as a Markov decision process defined by a tuple $\{\mathcal{S}, \mathcal{A}, \mathcal{T}, R, \rho, \gamma\}$ of states, actions, transition probability, reward, initial state distribution, and discount factor. An action distribution of an agent is represented as a policy $\pi_\theta(a_t|s_t)$, where $s_t \in \mathcal{S}$ is a state, $a_t \in \mathcal{A}$ is an action at time $t$, and $\theta$ are the parameters of the policy. An initial state $s_0$ is randomly sampled from $\rho$, and then, an agent iteratively takes an action $a_t$ sampled from a policy $\pi_\theta(a_t|s_t)$ and receives a reward $r_t$ until the episode ends. The performance of the agent is evaluated based on a discounted return $R = \sum_{t=0}^{T-1} \gamma^t r_t$, where $T$ is the episode horizon.

### 3.2 MODULAR FRAMEWORK WITH TRANSITION POLICIES

To learn a new task given primitive skills $\{p_1, p_2, \ldots, p_n\}$, we design a modular framework that consists of the following components: a meta-policy, primitive policies, and transition policies. The meta-policy chooses a primitive skill $p_c$ to execute at the beginning and whenever the primitive skill is terminated. Prior to running $p_c$, the transition policy for $p_c$ is executed to bring the current state to a plausible initial state for $p_c$, and therefore, $p_c$ can be successfully performed. This procedure is repeated to compose complex skills as illustrated in Figure 2 and Algorithm 2.

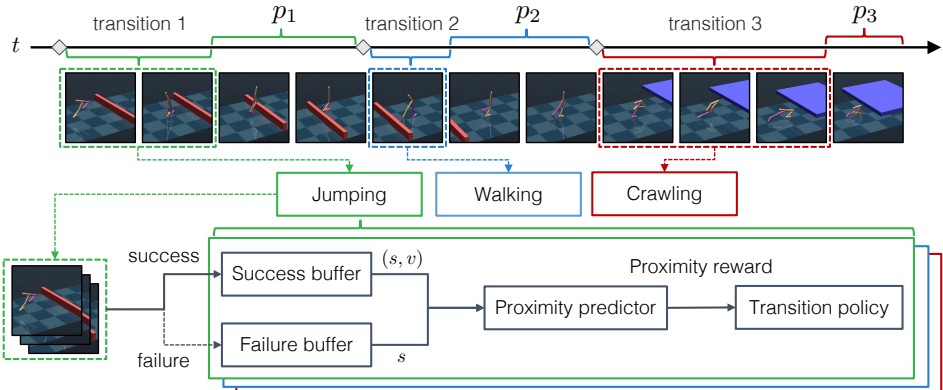

Figure 3: Training of transition policies and proximity predictors. After executing a primitive policy, a previously performed transition trajectory is labeled and added to a replay buffer based on the execution success. A proximity predictor is trained on states sampled from the two buffers to output the proximity to the initiation set. The predicted proximity serves as a reward to encourage the transition policy to move toward good initial states for the corresponding primitive policy.

We denote the meta-policy as $\pi_{meta}(p_c|s)$, where $c \in [1, n]$ is a primitive policy index. The observation of the meta-policy contains the low-level information of primitives and task specifications indicating high-level goals (e.g. moving direction and target object position). For example, a walking primitive only takes joint information as observation while the meta-policy additionally takes target direction. In this paper, we use a rule-based meta-policy and focus on transitioning between consecutive primitive policies.

Once a primitive skill $p_c$ is chosen to be executed, the agent generates an action $a_t \sim \pi_{p_c}(a|s_t)$ based on the current state $s_t$. Note that we did not differentiate state spaces for primitive polices because of the simplicity of notations (e.g. the observation of the jumping primitive contains a distance to a curb while that of the walking primitive only has joint pose and velocities). Every primitive policy is required to generate termination signals $\tau_{p_c} \in \{\text{continue}, \text{success}, \text{fail}\}$ to indicate policy completion and whether it believes the execution is successful or not. While our method is agnostic to the form of primitive policies (e.g. rule-based, inverse kinematics), we consider the case of a pre-trained neural network in this paper.

For smooth transitions between primitive policies, we add a transition policy $\pi_{\phi_c}(a|s)$ before executing primitive skill $p_c$, which guides an agent to $p_c$'s initiation set, where $\phi_c$ is the parameters of the transition policy for $p_c$. Note that the transition policy for $p_c$ is shared across different preceding primitive policies since a successful transition is defined by the success of the following primitive skill $p_c$. For brevity of notation, we omit the primitive policy index $c$ in the following equations where unambiguous. The transition policy's state and action space are the same as the primitive policy's. The transition policy also learns a termination signal $\tau_{\text{trans}}$ which indicates transition termination to successfully initiate $p_c$. Our framework contains one transition policy for each primitive skill, in total $n$ transition policies $\{\pi_{\phi_1}, \pi_{\phi_2}, \dots, \pi_{\phi_n}\}$.

### 3.3 TRAINING TRANSITION POLICIES

In our framework, transition policies are trained to make the execution of the corresponding following primitive policies successful. During rollouts, transition trajectories are collected and each trajectory can be naively labeled by the success execution of its corresponding primitive policy. Then, transition policies are trained to maximize the average success of the respective primitive policy. In this scenario, by definition, the only available learning signal for the transition policies is the sparse and binary rewards for the completion of the next task.

To alleviate the sparsity of rewards and maximize the objective of moving to viable initial states for the next primitive, we propose a *proximity predictor* that learns and provides a dense reward, dubbed *proximity reward*, of how close transition states are to the initiation set of the corresponding primitive $p_c$ as shown in Figure 3. We denote a proximity predictor as $P_{\omega_c}$ which is parameterized

by $\omega_c$. We define the proximity of a state as the future discounted proximity, $v = \delta^{step}$, where $step$ is the number of steps required to reach an initiation set of the following primitive policy. The proximity of a state can also be a linearly discounted function such as $v = 1 - \delta \cdot step$. We refer the readers to the supplementary for comparison of two proximity functions.

The proximity predictor is trained to minimize a mean squared error of proximity prediction:

$$L_P(\omega, \mathcal{B}^S, \mathcal{B}^F) = \frac{1}{2}\mathbb{E}_{(s,v)\sim\mathcal{B}^S}[(P_\omega(s) - v)^2] + \frac{1}{2}\mathbb{E}_{s\sim\mathcal{B}^F}[P_\omega(s)^2], \tag{1}$$

where $\mathcal{B}^S$ and $\mathcal{B}^F$ are collections of states from success and failure transition trajectories, respectively. To estimate the proximity to an initiation set, $\mathcal{B}^S$ contains not only the state that directly leads to the success of the following primitive policy, but also the intermediate states of the successful trajectories with its proximity. By minimizing this objective, given a state, the proximity predictor is learned to predict 1 if the state is in the initiation set, a value that is between 0 and 1 if the state leads the agent to end up with a desired initial states, and 0 when the state leads to a failure.

The goal of a transition policy is to get close to an initiation set which can be formulated as seeking a state $s$ predicted to be in the initiation set by the proximity predictor (i.e. $P_\omega(s)$ is close to 1). To achieve this goal, the transition policy learns to maximize proximity prediction at the ending state of the transition trajectory $P_\omega(s_T)$. In addition to providing reward at the end, we also use the increase of predicted proximity to the initiation set, $P_\omega(s_{t+1}) - P_\omega(s_t)$, at every timestep as a reward, dubbed *proximity reward*, to create a denser reward. The transition policy is trained to maximize the expected discounted return:

$$R_{\text{trans}}(\phi) = \mathbb{E}_{(s_0,s_1,\ldots,s_T)\sim\pi_\phi}\left[\gamma^T P_\omega(s_T) + \sum_{t=0}^{T-1}\gamma^t(P_\omega(s_{t+1}) - P_\omega(s_t))\right]. \tag{2}$$

However, in general skill learning scenarios, ground truth states ($\mathcal{B}^S$ and $\mathcal{B}^F$) for training proximity predictors are not available. Hence, the training data for a proximity predictor is obtained online during training its corresponding transition policy. Specifically, we label the states in a transition trajectory as success or failure based on whether the following primitive is successfully executed or not, and add them into the corresponding buffers $\mathcal{B}^S$ or $\mathcal{B}^F$, respectively. As stated in Algorithm 1, we train transition policies and proximity predictors by alternating between an Adam (Kingma & Ba, 2015) gradient step on $\omega$ to minimize Equation (1) with respect to $P_\omega$ and a PPO (Schulman et al., 2017) step on $\phi$ to maximize Equation (2) with respect to $\pi_\phi$. We refer readers to the supplementary for further details on training.

In summary, we propose to compose complex skills with transition policies that enable smooth transition between previously acquired primitive policies. Specifically, we propose to reward transition policies based on how close the current state is to suitable initial states of the subsequent policy (i.e. initiation set). To provide the proximity of a state, we collect failing and successful trajectories on the fly and train a proximity predictor to predict the proximity.

Utilizing the learned proximity predictors and proximity rewards for training transition policies is beneficial in the following perspectives: (1) the dense rewards speed up transition policy training by differentiating failing states from states in a successful trajectory; and (2) the joint training mechanism prevents a transition policy from getting stuck in local optima. Whenever a transition policy gets into a local optimum (i.e. fails the following skill with a high proximity reward), the proximity predictor learns to lower the proximity for the failing transition as those states are added to its failure buffer, escaping the local optimum.

## 4 EXPERIMENTS

We conducted experiments on two classes of continuous control tasks: robotic manipulation and locomotion. To illustrate the potential of the proposed framework, modular framework with Transition Policies (TP), we designed a set of complex tasks that require agents to utilize diverse primitive skills which are not optimized for smooth composition. All of our environments are simulated in the MuJoCo physics engine (Todorov et al., 2012).

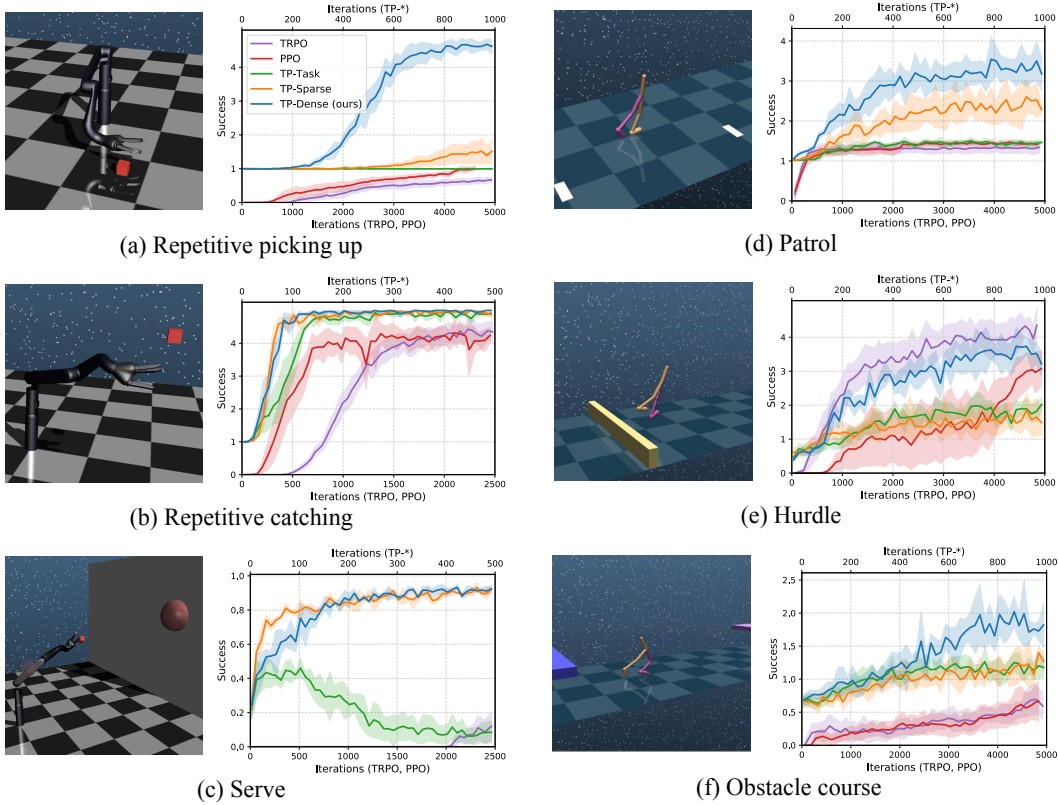

Figure 4: Tasks and success count curves of our model (blue), TRPO (purple), PPO (magenta), and transition policies (TP) trained on task reward (green) and sparse proximity reward (yellow). Our model achieves the best performance and convergence time. Note that TRPO and PPO are trained 5 times longer than ours with dense rewards since TRPO and PPO do not have primitive skills and learn from scratch. In the success count curves, different temporal scales are used for TRPO and PPO (bottom x-axis) and ours (top x-axis).

## 4.1 BASELINES

We evaluate our method to answer how transition policies benefit complex task learning and how joint training with proximity predictors boosts training of transition policies. To investigate the impact of the transition policy, we compared policies learned from dense rewards with our modular framework that only learns from sparse and binary rewards (i.e. subtask completion rewards). Moreover, we conducted ablation studies to dissect each component in the training method of transition polices. To answer these questions, we compare the following methods:

- **Trust Region Policy Optimization with dense reward (TRPO)** represents a state-of-the-art policy gradient method (Schulman et al., 2015), which we use for the standard RL comparison.

- **Proximal Policy Optimization with dense reward (PPO)** is another state-of-the-art policy gradient method (Schulman et al., 2017), which is more stable than TRPO with smaller batch sizes.

- **Without transition policies (Without-TP)** sequentially executes primitive policies without transition policies and has no learnable components.

- **Transition policies trained on task rewards (TP-Task)** represents a modular network augmented with transition policies learned from the sparse and binary reward (i.e. subtask completion reward), whereas our model learns from the dense proximity reward.

- **Transition policies trained on sparse proximity rewards (TP-Sparse)** is a variant of our model which has the proximity reward only at the end of the transition trajectory. In contrast, our model learns from dense proximity rewards generated every timestep.

Table 1: Success count for robotic manipulation, comparing our method against baselines with or without transition policies (TP). Our method achieves the best performance over both RL baselines and the ablated variants. Each entry in the table represents average success count and standard deviation over 50 runs with 3 random seeds.

|  | Reward | Repetitive picking up | Repetitive catching | Serve |
|---|---|---|---|---|
| TRPO | dense | $0.69 \pm 0.46$ | $4.54 \pm 1.21$ | $0.32 \pm 0.47$ |
| PPO | dense | $0.95 \pm 0.53$ | $4.26 \pm 1.63$ | $0.00 \pm 0.00$ |
| Without TP | sparse | $0.99 \pm 0.08$ | $1.00 \pm 0.00$ | $0.11 \pm 0.32$ |
| TP-Task | sparse | $0.99 \pm 0.08$ | $4.87 \pm 0.58$ | $0.05 \pm 0.21$ |
| TP-Sparse | sparse | $1.52 \pm 1.12$ | $4.88 \pm 0.59$ | $\mathbf{0.92 \pm 0.27}$ |
| TP-Dense (ours) | sparse | $\mathbf{4.84 \pm 0.63}$ | $\mathbf{4.97 \pm 0.33}$ | $\mathbf{0.92 \pm 0.27}$ |

- **Transition policies trained on dense proximity rewards (TP-Dense, Ours)** is our final model where transition policies learn from dense proximity rewards.

Initially, we tried comparing baseline methods with our method using only sparse and binary rewards. However, the baselines could not solve any of the tasks due to the complexity and sparse reward of the environments. To provide more competitive comparisons, we engineer dense rewards for baselines (TRPO and PPO) to boost their performance and give baselines 5 times longer training times. We show that transitions with sparse rewards can compete with and even outperform baselines learning from dense rewards. As the performance of TRPO and PPO varies significantly between runs, we train each task with 3 different random seeds and report mean and standard deviation in Figure 4.

## 4.2 ROBOTIC MANIPULATION

For robotic manipulation, we simulate a Kinova Jaco, a 9 DoF robotic arm with 3 fingers. The agent receives full state information, including the absolute location of external objects. The agent uses joint torque control to perform actions. The results are shown in Figure 4 and Table 1.

**Pre-trained primitives.** There are four pre-trained primitives available: *Picking up*, *Catching*, *Tossing*, and *Hitting*. *Picking up* requires the robotic arm to pick up a small block, which is randomly placed on the table. If the box is not picked up after a certain amount of time, the agent fails. *Catching* learns to catch a block that is thrown towards the arm with random initial position and velocity. The agent fails if it does not catch and stably hold the box for a certain amount of time. *Tossing* requires the robot to pick up a box, toss it vertically in the air, and land the box at a specified position. *Hitting* requires the robot to hit a box dropped overhead at a target ball.

**Repetitive picking up.** The *Repetitive picking up* task requires the agent to complete the *Picking up* task 5 times. After each successful pick, the box disappears and a new box will be placed randomly on the table again. Our model achieves the best performance and converges the fastest by learning from the proposed proximity reward. With our dense proximity reward at every transition step, we alleviate credit assignment when compared to providing a sparse proximity reward (TP-Sparse) or using a sparse task reward (TP-Task). Conversely, TRPO and PPO with dense rewards take significantly longer to learn and is unable to pick up the second box as the ending pose after the first picking up is too unstable to initialize the next picking up.

**Repetitive catching.** Similar to *Repetitive picking up*, the *Repetitive catching* task requires the agent to catch boxes consecutively up to 5 times. In this task, other than the modular network without a transition policy, all baselines are able to eventually learn while our model still learns the fastest. We believe this is because the *Catching* primitive policy has a larger initiation set and therefore, the sparse reward problem is less severe since random exploration is able to succeed with a higher chance.

**Serve.** Inspired by tennis, *Serve* requires the robot to toss the ball and hit it at a target. Even with an extensively engineered reward, TRPO and PPO baselines fail to learn because *Hitting* is not able to learn to cover all terminal states of *Tossing* (i.e. a set of initial states for *Hitting* is large which demands longer training time). In contrast, learning to recover from *Tossing*'s ending states to *Hitting*'s initiation set is easier for exploration (11% of *Tossing*'s ending states are covered by

Table 2: Success count for locomotion, comparing our method against baselines with or without transition policies (TP). Our method outperforms all baselines in *Patrol* and *Obstacle course*. In *Hurdle*, the reward function for TRPO was extensively engineered, which is not directly comparable to our method. Our method outperforms baselines learning from sparse reward, showing the effectiveness of the proposed proximity predictor. Each entry in the table represents average success count and standard deviation over 50 runs with 3 random seeds.

|                | Reward | Patrol | Hurdle | Obstacle course |
|----------------|--------|--------|--------|-----------------|
| TRPO           | dense  | $1.37 \pm 0.52$ | $\mathbf{4.13 \pm 1.54}$ | $0.98 \pm 1.09$ |
| PPO            | dense  | $1.53 \pm 0.53$ | $2.87 \pm 1.92$ | $0.85 \pm 1.07$ |
| Without TP     | sparse | $1.02 \pm 0.14$ | $0.49 \pm 0.75$ | $0.72 \pm 0.72$ |
| TP-Task        | sparse | $1.69 \pm 0.63$ | $1.73 \pm 1.28$ | $1.08 \pm 0.78$ |
| TP-Sparse      | sparse | $2.51 \pm 1.26$ | $1.47 \pm 1.53$ | $1.32 \pm 0.99$ |
| TP-Dense (Ours)| sparse | $\mathbf{3.33 \pm 1.38}$ | $\mathbf{3.14 \pm 1.69}*$ | $\mathbf{1.90 \pm 1.45}$ |

*Hitting*'s initiation set as can be seen in Table 1), which reduces the complexity of the task. Thus, our method and the sparse proximity reward baseline are both able to solve it. However, the ablated variant trained on task reward shows high success rates at the beginning of training and collapses after 100 iterations. The performance drops because the transition policy tries to solve failure cases by increasing the transition length and it reaches to a point that it hardly gets reward. This result shows that once the policy falls into local optima, it is not able to escape because the policy will never get a sparse task reward. On the other hand, our method is robust to local optima since the jointly learned dense proximity reward provides a learning signal to an agent even though it cannot get a task reward.

## 4.3 LOCOMOTION

For locomotion, we simulate a 9 DoF planar (2D) bipedal walker. The observation of the agent includes joint position, rotation, and velocity. When the agent needs to interact with objects in the environment, we provide additional input such as distance to the curb and ceiling in front of the agent. The agent uses joint torque control to perform actions. The results are shown in Figure 4 and Table 2.

**Pre-trained primitives.** *Forward* and *Backward* require the walker to walk forward and backward with a certain velocity, respectively. *Balancing* requires the walker to robustly stand still under the random external forces. *Jumping* requires the walker jump over a randomly located curb and land safely. *Crawling* requires the walker to crawl under a ceiling. In all the aforementioned scenarios, the walker fails when the height of the walker is lower than a threshold.

**Patrol (Forward and backward).** The *Patrol* task involves walking forward and backward toward goal points on either side and balancing in between to smoothly change its direction. As illustrated in Figure 4, our method consistently outperforms TRPO, PPO, and ablated baselines in stably walking forward and transitioning to walk backward. The agent trained with dense rewards is not able to consistently switch directions, whereas our model can utilize previously learned primitives including *Balancing* to stabilize a reversal in velocity.

**Hurdle (Walking forward and jumping).** The *Hurdle* task requires the agent to walk forward and jump across curbs, which requires a transition between walking and jumping as well as landing the jump to walking forward. As shown in Figure 4, our method outperforms the sparse reward baselines, showing the efficiency our proposed proximity reward. While TRPO with dense rewards can learn this task as well, it requires dense rewards consisting of eight different components to collectively enable TRPO to learn the task. It can be considered as learning both primitive skills and transition between skills from dense rewards. However, the main focus of this paper is to learn a complex task by reusing acquired skills, avoiding an extensive reward design.

**Obstacle Course (Walking forward, jumping, and crawling).** *Obstacle Course* is the most difficult among the locomotion tasks, where the walker must walk forward, jump across curbs, and crawl underneath ceilings. It requires three different behaviors and transitions between two very different primitive skills: crawling and jumping. Since the task requires significantly different behaviors that are hard to transition between, TRPO fails to learn the task and only tries to crawl toward the curb

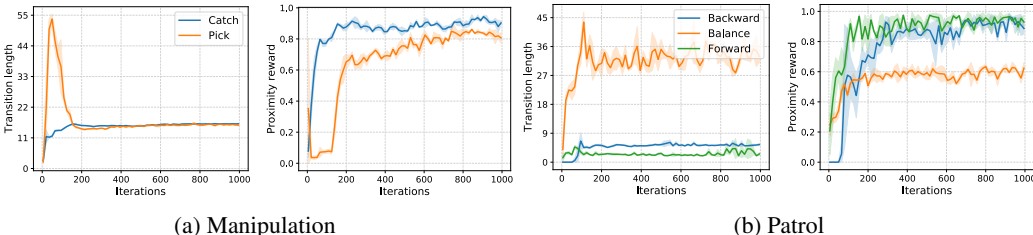

(a) Manipulation                (b) Patrol

Figure 5: Average transition length and average proximity reward of transition trajectories over training on *Manipulation* (left) and *Patrol* (right).

without attempting to jump. In contrast, our method learns to transition between all pairs of primitive skills and often succeeds in crossing multiple obstacles.

## 4.4 ABLATION STUDY

We conducted additional experiments to understand the contribution of transition policies, proximity predictors, and dense proximity rewards. The modular framework without transition policies (Without-TP) tends to fail the execution of the second skill since the second skill is not trained to cover ending states of the first skill. Especially, in continuous control making a primitive skill that can cover all possible states is very challenging. Transition policies trained from task completion reward (TP-Task) and sparse proximity reward (TP-Sparse) learn to connect consecutive primitives slower because sparse reward is hard to learn from due to the credit assignment problem. On the other hand, our model alleviates the credit assignment problem and learns quickly by giving predicted proximity reward for every transition state-action pair.

## 4.5 TRAINING OF TRANSITION AND PROXIMITY PREDICTOR

To investigate how transition polices learn to solve the tasks, we present the lengths of transition trajectories and the obtained proximity rewards during training in Figure 5. For manipulation, we show the results of *Repetitive picking up* and *Repetitive catching*. For locomotion, we show *Patrol* with three different transition policies.

The transition policy quickly learns to maximize the proximity reward regardless of the accuracy of the proximity predictor. All the transition policies increase the length while exploring in the beginning, especially for *picking up* (55 steps) and *balance* (45 steps). This is because a randomly initialized proximity predictor outputs high proximity for unseen states and a transition policy tries to get a high reward by visiting these states. However, as these failing initial states with high proximity are collected in the failure buffers, the proximity predictor lowers their proximity and the transition policy learns to avoid them. In other words, the transition policy will end up seeking successful states. As transition policies learn to transition to the following skills, the length decreases to get higher proximity rewards earlier.

## 4.6 VISUALIZING TRANSITION TRAJECTORY

Figure 6a shows two transition trajectories (from $s_0$ to $t_0$ and $s_1$ to $t_1$) and two-dimensional PCA embedding of the ending states (blue) and initiation states (red) of the *Picking up* primitive. A transition policy starts from states $s_0$ and $s_1$ where the previous *Picking up* primitive is terminated. As can be seen in Figure 6a, the proximity predictor outputs small values for $s_0$ and $s_1$ since they are far from the initiation set of *Picking up* primitive. Trajectories in the figure show that as the transition policy moves toward states with higher proximity, and finally ends up with states $t_0$ and $t_1$ which are in the initiation set of the primitive policy.

Figure 6b illustrates PCA embeddings of initiation sets of three primitive skills, *Forward* (green), *Backward* (orange), and *Balancing* (blue). A transition from *Forward* to *Balancing* has very long trajectory, but predicted proximity helps the transition policy to reach to an initiation state $t_0$. On the other hand, transitioning between *Balancing* and *Backward* only requires 7 steps.

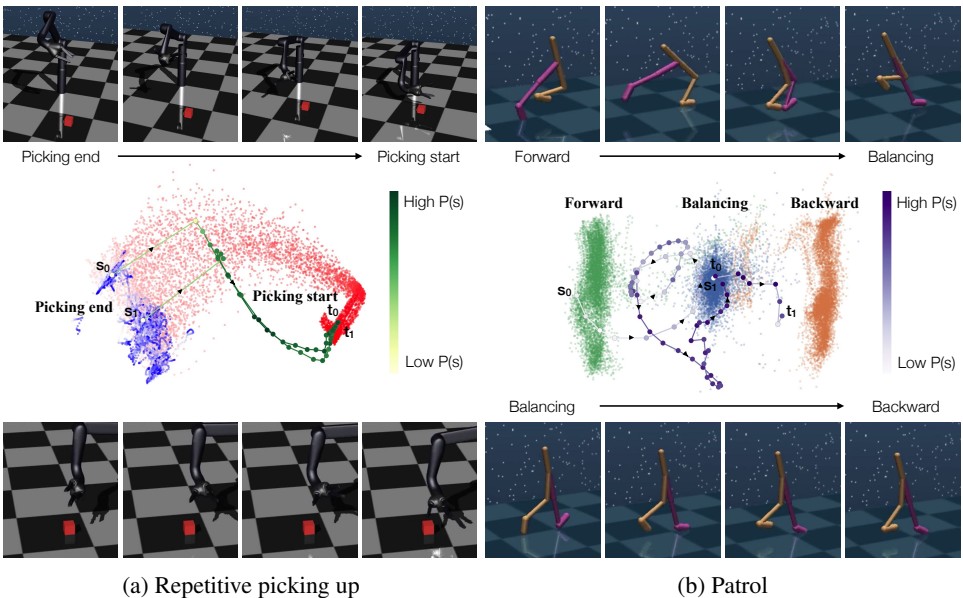

(a) Repetitive picking up            (b) Patrol

Figure 6: Visualization of transition trajectories of (a) *Repetitive picking up* and (b) *Patrol*. TOP AND BOTTOM ROWS: contain rendered frames of transition trajectories. MIDDLE ROW: contains states extracted from each primitive skill execution projected onto PCA space. The dots connected with lines are extracted from the same transition trajectory, where the marker color indicates the proximity prediction $P(s)$. A higher $P(s)$ value indicates proximity to states suitable for initializing the next primitive skill. LEFT: two picking up transition trajectories demonstrate that the transition policy learns to navigate from terminate states $s_0$ and $s_1$ to $t_0$ and $t_1$. RIGHT: the forward to balance transition moves between the forward and balance state distributions and the balance to backward transition moves from the balancing states close to the backward states.

## 5   CONCLUSION

In this work, we propose a modular framework with transition policies to empower reinforcement learning agents to learn complex tasks with sparse reward by utilizing prior knowledge. Specifically, we formulate the problem as executing existing primitive skills while smoothly transitioning between primitive skills. To learn transition polices in a sparse reward setting, we propose a proximity predictor which generates dense reward signals and jointly train transition policies and proximity predictors. Our experimental results on robotic manipulation and locomotion tasks demonstrate the effectiveness of employing transition policies. The proposed framework solves complex tasks without reward shaping and outperforms baseline RL algorithms and other ablated baselines.

There are many future directions to investigate. Our method is designed to focus on acquiring transition policies that connect a given set of primitive policies under the predefined meta-policy. We believe that joint learning of a meta-policy and transition policies on a new task would make our framework more flexible. Moreover, we made an assumption that successful transition between two consecutive policies should be achievable by random exploration. To alleviate the exploration problem with sparse rewards, our transition policy training can incorporate exploration methods such as count-based exploration bonuses (Bellemare et al., 2016; Martin et al., 2017) and curiosity-driven intrinsic reward (Pathak et al., 2017). We also assume our primitive policies return a signal that indicates whether the execution should be terminated or not, similar to Kulkarni et al. (2016); Oh et al. (2017); Le et al. (2018). Learning to assess the successful termination of primitive policies together with learning transition policies is a promising future direction.

### ACKNOWLEDGMENTS

This project was supported by the center for super intelligence, Kakao Brain, and SKT. The authors would like to thank Yuan-Hong Liao for helpful discussions during initial ideation.

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

## A    ACQUIRING PRIMITIVE POLICIES

The modular framework proposed in this paper allows a primitive policy to be any of a pre-trained neural network, inverse kinematics module, or hard-coded policy. In this paper, we use neural networks trained with TRPO (Schulman et al., 2015) on dedicated environments as primitive policies (see Section C for the details of environments and reward functions). All policy networks we used consists of 2 layers of 32 hidden units with $\tanh$ nonlinearities and predicts the mean and standard deviation of a Gaussian distribution over an action space. We trained all primitive policies until the total return converged (up to 10,000 iterations).

Given a state, a primitive policy outputs an *action* as well as a *termination signal* indicating whether the execution is done and if the skill was successfully performed (see Section C for details on primitive skills and termination conditions).

## B    TRAINING DETAILS

### B.1    IMPLEMENTATION DETAILS

For the TRPO and PPO implementation, we used OpenAI baselines (Dhariwal et al., 2017) with default hyperparameters including learning rate, KL penalty, and entropy coefficients unless specified below.

| Hyperparameters | Transition policy | Proximity predictor | Primitive policy | TRPO | PPO |
|---|---|---|---|---|---|
| Learning rate | 1e-4 | 1e-4 | 1e-3 (for critic) | 1e-3 (for critic) | 1e-4 |
| # Mini-batch | 150 | 150 | 32 | 150 | 150 |
| Mini-batch size | 64 | 64 | 64 | 64 | 64 |
| Learning rate decay | no | no | no | no | linear decay |

Table 3: Hyperparameter values for transition policy, proximity predictor, and primitive policy as well as TRPO and PPO baselines.

For all networks, we use the Adam optimizer with mini-batch size of 64. We use 4 workers for rollout and parameter update. The size of rollout for each update is 10,000 steps. We limit the maximum length of a transition trajectory as 100.

### B.2    REPLAY BUFFERS

A success buffer $\mathcal{B}^S$ contains states and their proximity to the corresponding initiation set in successful transitions. On the other hand, a failure buffer $\mathcal{B}^F$ contains states in failure transitions. Both the two buffers are FIFO (i.e. new items are added on one end and once a buffer is full, a corresponding number of items are discarded from the opposite end). For all experiments, we use buffers, $\mathcal{B}^S$ and $\mathcal{B}^F$, with a capacity of one million states.

For efficient training of the proximity predictors, we collect successful trajectories of primitive skills which can be sampled during the training of primitive skills. We run 1,000 episodes for each primitive and put the first 10 - 20% in trajectories into the success buffer as an initiation set. While initiation sets can be discovered via random exploration, we found that this initialization of success buffers improves the efficiency of training by providing initial training data for the proximity predictors.

### B.3    PROXIMITY REWARD

Transition policies receive rewards based on the outputs of proximity predictors. Before computing the reward at every time step, we clip the output of the proximity predictor $P$ by $\text{clip}(P(s), 0, 1)$ which indicates how close the state $s$ is to the initiation set of the following primitive (higher values correspond to closer states). We define the proximity of a state to an initiation set as an exponentially discounted function $\delta^{step}$, where $step$ is the shortest number of timesteps required to get to a state in the initiation set. We use $\delta = 0.95$ for all experiments. To make the reward denser, for every timestep $t$, we provide the increase in proximity, $P(s_{t+1}) - P(s_t)$, as a reward for transition policy.

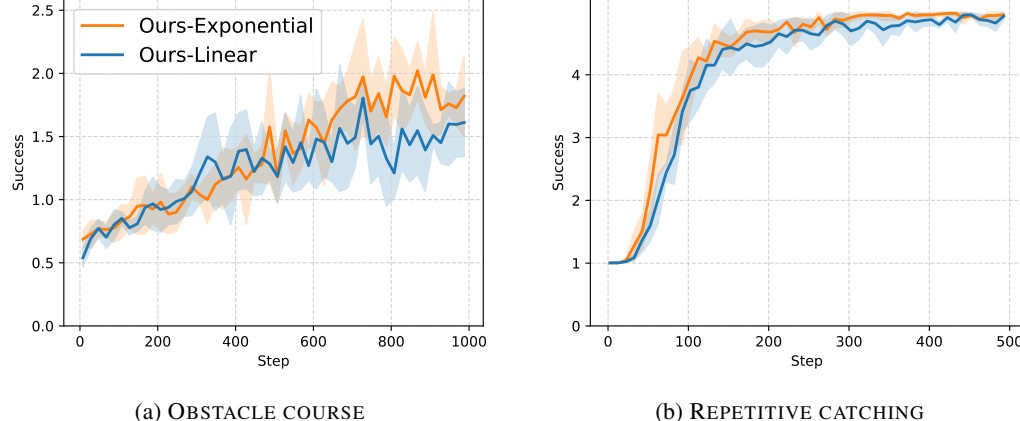

(a) OBSTACLE COURSE  (b) REPETITIVE CATCHING

Figure 7: Success count curves of our model with exponentially discounted proximity function and linearly discounted proximity function over training on *Obstacle course* (left) and *Repetitive catching* (right).

Using a linearly discounted proximity function, $1 - \delta \cdot step$, is also a valid choice. We compare the two proximity functions on a manipulation task (*Repetitive catching*) and a locomotion task (*Obstacle course*), as shown in Figure 7, where $\delta$ for exponential decay and linear decay are $0.95$ and $0.01$, respectively. The results demonstrate that our model is able to learn well with both proximity functions and they perform similarly.

Originally, we opted for the exponential proximity function with the intuition that the faster initial decay near the initiation set would help the policy discriminate successful states from failing states near the initiation set. Also, in our experiments, as we use $0.95$ as a decaying factor, the proximity is still reasonably large (e.g., $0.35$ for 20 time-steps and $0.07$ for 50 time-steps). In this paper, we use the exponential proximity function for all experiments.

## B.4 PROXIMITY PREDICTOR

A proximity predictor takes a state as input which includes joint state information, joint acceleration, and any task specification, such as ceiling and curb information. A proximity predictor consists of 2 fully connected layers of 96 hidden units with ReLU nonlinearities and predicts the proximity to the initiation set based on the states sampled from the success and failure buffers. Each training iteration consists of 10 epochs over a batch size of 64 and use a learning rate of $10^{-4}$. The predictor optimizes the loss in Equation (1), similar to the LSGAN loss (Mao et al., 2017).

## B.5 TRANSITION POLICIES

An observation space of a transition policy consists of joint state information and joint acceleration. A transition policy consists of 2 fully connected layers of 32 hidden units with $\tanh$ nonlinearities and predicts the mean and standard deviation of a Gaussian distribution over an action space. A 2-way softmax layer is followed by the last fully connected layer to predict whether to terminate the current transition or not. We train all transition policies using PPO (Schulman et al., 2017) since PPO is robust on smaller batch sizes and the transition states collected for each update is much smaller than the size of a rollout. Each training iteration consists of 5 epochs over a batch.

---

**Algorithm 1** TRAIN

---

1: **Input:** Primitive polices $\{\pi_{p_1}, ..., \pi_{p_n}\}$.
2: Initialize success buffers $\{\mathcal{B}_1^S, ..., \mathcal{B}_n^S\}$ with successful trajectories of primitive policies.
3: Initialize failure buffers $\{\mathcal{B}_1^F, ..., \mathcal{B}_n^F\}$.
4: Randomly initialize parameters of transition policies $\{\phi_1, ..., \phi_n\}$ and proximity predictors $\{\omega_1, ..., \omega_n\}$.
5: **repeat**
6:     Initialize rollout buffers $\{\mathcal{R}_1, ..., \mathcal{R}_n\}$.
7:     Collect trajectories using ROLLOUT.
8:     **for** $i = 1$ **to** $n$ **do**
9:         Update $P_{\omega_i}$ to minimize Equation (1) using $\mathcal{B}_i^S$ and $\mathcal{B}_i^F$.
10:        Update $\pi_{\phi_i}$ to maximize Equation (2) using $\mathcal{R}_i$.
11:    **end for**
12: **until** convergence

---

**Algorithm 2** ROLLOUT

---

1: **Input:** Meta policy $\pi_{\text{meta}}$, primitive policies $\{\pi_{p_1}, ..., \pi_{p_n}\}$, transition policies $\{\pi_{\phi_1}, ..., \pi_{\phi_n}\}$, and proximity predictors $\{P_{\omega_1}, ..., P_{\omega_n}\}$.
2: Initialize an episode and receive initial state $s_0$.
3: $t \leftarrow 0$
4: **while** episode is not terminated **do**
5:     $c \sim \pi_{\text{meta}}(s_t)$
6:     Initialize a rollout buffer $\mathcal{B}$.
7:     **while** episode is not terminated **do**
8:         $a_t, \tau_{\text{trans}} \sim \pi_{\phi_c}(s_t)$
9:         Terminate the transition policy if $\tau_{\text{trans}} = \text{terminate}$.
10:        $s_{t+1}, \tau_{\text{env}} \leftarrow \text{ENV}(s_t, a_t)$
11:        $r_t \leftarrow P_{\omega_c}(s_{t+1}) - P_{\omega_c}(s_t)$
12:        Store $(s_t, a_t, r_t, \tau_{\text{env}}, s_{t+1})$ in $\mathcal{B}$
13:        $t \leftarrow t + 1$
14:    **end while**
15:    **while** episode is not terminated **do**
16:        $a_t, \tau_{p_c} \sim \pi_{p_c}(s_t)$
17:        Terminate the primitive policy if $\tau_{p_c} \neq \text{continue}$.
18:        $s_{t+1}, \tau_{\text{env}} \leftarrow \text{ENV}(s_t, a_t)$
19:        $t \leftarrow t + 1$
20:    **end while**
21:    Compute the discounted proximity $v$ of each state $s$ in $\mathcal{B}$.
22:    Add pairs of $(s, v)$ to $\mathcal{B}_c^S$ or $\mathcal{B}_c^F$ according to $\tau_{p_c}$.
23:    Add $\mathcal{B}$ to the rollout buffer $\mathcal{R}_c$.
24: **end while**

## B.6 SCALABILITY

Each sub-policy requires its corresponding transition policy, proximity predictor, and two buffers. Hence, both the time and memory complexities of our method are linearly dependent on the number of sub-policies. The memory overhead is affordable since a transition policy (2 layers of 32 hidden units), a proximity predictor (2 layers of 96 hidden units), and replay buffers (1M states) are small.

## C ENVIRONMENT DESCRIPTIONS

For every task, we add a control penalty, $-0.001 * \|a\|^2$, to regularize the magnitude of actions where $a$ is a torque action performed by an agent. Note that all measures are in meters, and we omit the measures here for clarity of the presentation.

## C.1 ROBOTIC MANIPULATION

In object manipulation tasks, a 9-DOF Jaco robotic arm[1] is used as an agent and a cube with the side length 0.06 m is used as a target object. We follow the tasks and environment settings proposed in Ghosh et al. (2018). The observation consists of the position of the base of the Jaco arm, joint angles, angular velocities as well as the position, rotation, velocity, and angular velocity of the cube. The action space is a torque control on 9 joints.

### C.1.1 REWARD DESIGN AND TERMINATION CONDITION

**Picking up:** In the *Picking up* task, the position of the box is randomly initialized within a square region of size 0.1 m × 0.1 m with a center (0.5, 0.2). There is an initial guide reward to guide the arm to the box. There is also an over reward to guide the hand directly over the box. When the arm is not picking up the box, there is a pick reward to incentivize the arm to pick the box up. There is an additional hold reward that makes the arm hold the box in place after picking up. Finally, there is a success reward given after the arm has held the box for 50 frames. The success reward is scaled with number of timesteps to encourage the arm to succeed as quickly as possible.

$$R(s) = \lambda_{guide} \cdot \mathbf{1}_{\text{Box not picked and Box on ground}} + \lambda_{pick} \cdot \mathbf{1}_{\text{Box in hand and not picked}} + \lambda_{hold} \cdot \mathbf{1}_{\text{Box picked and near hold point}}$$

$$\lambda_{guide} = 2, \lambda_{pick} = 100, \lambda_{hold} = 0.1$$

**Catching:** The position of the box is initialized at (0, 2.0, 1.5) and the directional force of size 110 is applied to throw the box toward the agent with randomness (0.1 m × 0.1 m).

$$R(s) = \mathbf{1}_{\text{Box in air and Box within 0.06 of Jaco end-effector}}$$

**Tossing:** The box is randomly initialized on the ground at (0.4, 0.3, 0.05) within a 0.005 × 0.005 square region. A guide reward is given to guide the arm to the top of the box. A pick reward is then given to lift the box up to a specified release height. A release reward is given if the box is no longer in the hand. A stable reward is given to minimize variation in the box's x and y direction. An up reward is given while the ball is traveling upwards in air, up until the box hits a specified z height. Finally, a success reward +100 is given based on the landing position of the box and the specified landing position.

**Hitting:** The box is randomly initialized overhead the arm at (0.4, 0.3, 1.2) within a 0.005 × 0.005 m square region. The box falls and the arm is given a hit reward +10 for hitting the box. Once the box has been hit, a target reward is given based on how close the box is to the target.

**Repetitive picking up:** The *Repetitive picking up* task has two reward variants. The sparse version gives a reward +1 for every successful pick. The dense reward version gives a guide reward to the box after each successful pick following the reward for the *Picking up* task.

**Repetitive catching:** The *Repetitive catching* task gives a reward +1 for every successful catch. For dense reward, it uses the same reward function with that of the *Catching* task.

**Serve:** The *Serve* task gives a toss reward +1 for a successful toss and a target reward +1 for successfully hitting the target. The dense reward setting provides the *Tossing* and *Hitting* reward according to box position.

## C.2 LOCOMOTION

A 9-DOF bipedal planar walker is used for simulating locomotion tasks. The observation consists of the position and velocity of the torso, joint angles, and angular velocities. The action space is torque control on the 6 joints.

---

[1] http://www.mujoco.org/forum/index.php?resources/kinova-arms.12/

### C.2.1 REWARD DESIGN

Different locomotion tasks share many components of reward design, such as velocity, stability, and posture. We use the same form of reward functions, but with different hyperparameters for each task. The basic form of the reward function is as following:

$$R(s) = \lambda_{vel} \cdot \text{abs}(v_x - v_{target}) + \lambda_{alive} - \lambda_{height} \cdot \text{abs}(1.1 - min(1.1, \Delta h)) +$$
$$\lambda_{angle} \cdot \cos(angle) - \lambda_{foot}(v_{right\_foot} + v_{left\_foot}),$$

where $v_x$, $v_{right\_foot}$, and $v_{left\_foot}$ are forward velocity, right foot angular velocity, left foot angular velocity; and $\Delta h$ and $angle$ are the distance between the foot and torso and the angle of the torso, respectively. The foot velocities help the agent to move its feet naturally. $\Delta h$ and $angle$ are used to maintain height of the torso and encourage an upright pose.

**Forward:** The *Forward* task requires the walker agent to walk forward for 20 meters. To make the agent robust, we apply a random force with arbitrary magnitude and direction to a randomly selected joint every 10 timesteps.

$$\lambda_{vel} = 2, \lambda_{alive} = 1, \lambda_{height} = 2, \lambda_{angle} = 0.1, \lambda_{foot} = 0.01, \text{ and } v_{target} = 3$$

**Backward:** Similar to *Forward*, the *Backward* task requires the walker to walk backward for 20 meters under random forces.

$$\lambda_{vel} = 2, \lambda_{alive} = 1, \lambda_{height} = 2, \lambda_{angle} = 0.1, \lambda_{foot} = 0.01, \text{ and } v_{target} = -3$$

**Balancing:** In the *Balancing* task, the agent learns to balance under strong random forces for 1000 timesteps. Similar to other tasks, the random forces are applied to a random joint every 10 timesteps, but with magnitude 5 times larger.

$$\lambda_{vel} = 1, \lambda_{alive} = 1, \lambda_{height} = 0.5, \lambda_{angle} = 0.1, \lambda_{foot} = 0, \text{ and } v_{target} = 0$$

**Crawling:** In the *Crawling* task, a ceiling of height 1.0 and length 16 is located in front of the agent, and the agent is required to crawl under the ceiling without touching it. If the agent touches the ceiling, we terminate the episode. The task can be completed when the agent passes a point 1.5 after the ceiling and the agent gets 100 additional reward.

$$\lambda_{vel} = 2, \lambda_{alive} = 1, \lambda_{height} = 0, \lambda_{angle} = 0.1, \lambda_{foot} = 0.01, \text{ and } v_{target} = 3$$

**Jumping:** In the *Jumping* task, a curb of height 0.4 and length 0.2 is located in front of the walker agent. The observation contains a distance to the curb in addition to the 17-dimensional joint information, where the distance is clipped by 3. The x location of the curb is randomly chosen from [2.5, 5.5]. In addition to the reward function above, it also gets an additional 100 reward for passing the curb and $200 \cdot v_y$ when the agent passes the front, middle, and end slices of the curb, where $v_y$ is y-velocity. If the agent touches the curb, the agent gets -10 penalty and the episode is terminated.

$$\lambda_{vel} = 2, \lambda_{alive} = 1, \lambda_{height} = 2, \lambda_{angle} = 0.1, \lambda_{foot} = 0.01, \text{ and } v_{target} = 3$$

**Patrol:** The *Patrol* task is repetitive running forward and backward between two goals at $x = -2$ and $x = 2$. Once the agent touches a goal, the target is changed to another goal and the sparse reward +1 is given. The dense reward alternates between the reward functions of *Forward* and *Backward*. The agent gets the reward of *Forward* when the agent is heading toward $x = 2$ and gets the reward of *Backward*, otherwise.

**Hurdle:** The *Hurdle* environment consists of 5 curbs positioned at $x = \{8, 18, 28, 38, 48\}$ and requires repetitive walking and jumping behaviors. The position of each curb is randomized with a uniformly sampled value from $[-0.5, 0.5]$. The sparse reward +1 is given when the agent jumps over a curb (i.e. pass a point 1.5 after a curb).

The dense reward for *Hurdle* is same with *Jumping* and has 8 reward components to guide the agent to learn the desired behavior. By extensively designing dense rewards, it is possible to solve complex

tasks. In comparison, our proposed method learns from sparse reward by re-using prior knowledge and doesn't require reward shaping.

**Obstacle Course:** The *Obstacle Course* environment replaces two curbs in *Hurdle* with a ceiling of height 1.0 and length 3. The sparse reward +1 is given when the agent jumps over a curb or passes through a ceiling (i.e. pass a point 1.5 after a curb or a ceiling). The dense reward is alternating between *Jumping* before the curb and *Crawling* before the ceiling.

### C.2.2 TERMINATION SIGNAL

Locomotion tasks except *Crawling* fail if $h < 0.8$ and *Crawling* fails if $h < 0.3$. *Forward* and *Backward* tasks are considered as success when the walker reaches to the target or 5 in front of obstacles. *Balancing* task is considered successful when the agent does not fail for 50 timesteps. The agent succeeds on *Jumping* and *Crawling* if the agent passes the obstacles by a distance of 1.5.

