# OpenReview forum: "Composing Complex Skills by Learning Transition Policies"
_ICLR.cc/2019/Conference_

### Official Review · AnonReviewer2 · 2018-11-03
**Potentially very useful idea**

**Rating:** 7
**Confidence:** 4

**Review:**

** Summary **
The authors propose a new training scheme with a learned auxiliary reward function to optimise transition policies, i.e. policies that connect the ending state of a previous macro action/option with good initiation states of the following macro action/option.

** Quality & Clarity **
The paper is well written and features an extensive set of experiments.

** Originality **
I am not aware of similar work and believe the idea is novel.

** Significance **
Several recent papers have proposed to approach the topic of learning hierarchical policies not by training the hierarchy end-to-end, but by first learning useful individual behavioural patterns (e.g. skills) which then later can be used and sequentially chained together by higher-level policies. I believe the here presented work can be quite helpful to do so as the individual skills are not optimised for smooth composition and are therefore likely to fail when naively used sequentially.

---

> ### Author Response · Authors · 2018-11-12
> **Response to Reviewer 2**
>
> We thank the reviewer for the feedback. We are glad that the reviewer found the idea novel and useful for enabling the smooth composition of skills and that the reviewer recognized the importance of utilizing previously learned skills to compose complex skills.

---

### Official Review · AnonReviewer1 · 2018-11-05
**An elegant method with comprehensive evaluations**

**Rating:** 9
**Confidence:** 4

**Review:**

The paper presents a method for learning policies for transitioning from one task to another with the goal of completing complex tasks. In the heart of the method is state proximity estimator, which measures the distance between states in the originator and destination tasks. This estimator is used in the reward for the transition policy. The method is evaluated on number of MojoCo tasks, including locomotion and manipulation.

Strengths:
+ Well motivated and relevant topic. One of the big downsides in the current state of the art is lack of understanding how to learn complex tasks. This papers tackles that problem.
+ The paper is well written and the presentation is clear.
+ The method is simple, yet original. Overall, an elegant approach that appears to be working well.
+ Comprehensive evaluations over several tasks and several baselines.

Questions:
- In the metapolicy, what ensures consistency, i.e. it selects the same policy in the consecutive steps?
- Can the authors comment on the weaknesses and the limits of the method?

---

> ### Author Response · Authors · 2018-11-12
> **Response to Reviewer 1**
>
> We thank the reviewer for the feedback and address the concerns in detail below.
>
> > Reviewer 1 (R1): “In the metapolicy, what ensures consistency, … ?”
>
> Our meta-policy executes a primitive policy and waits for a termination signal from the primitive policy before choosing the subsequent one. In other words, a termination signal (success/failure of the primitive policy) comes from the primitive policy, i.e. the walker falls down or the arm picks up a box. This call-and-return style [1-3] of execution ensures the same policy is utilized in consecutive steps until its completion. Hierarchical reinforcement methods have employed this call-and-return style when sub-policies are learned for well-defined sub-tasks that do not require a context switch during their execution.
>
> > R1: “... the weaknesses and the limits of the method?”
>
> We discuss a few assumptions that we made and good follow-up directions below. We will also add the discussion to the revised version.
>
> Our model-free transition policies rely on random exploration. Specifically, we made an assumption that successful transition trajectories between two consecutive policies should be achievable by random exploration (i.e. an initiation set of a primitive policy should be reachable from the ending states of the previous policies). As soon as a transition policy succeeds once, the proximity predictor will learn what good states are and subsequently the transition policy will succeed more frequently. To alleviate the exploration problem with sparse rewards, our transition policy training can incorporate exploration methods that utilize count-based exploration bonuses [4-6], curiosity-driven intrinsic rewards [7-10], etc.
>
> Our current framework is designed to focus on acquiring transition policies that can connect a given set of primitive policies. We believe that additionally enabling an agent to adaptively augment its primitive set [11-12] based on a new environment or task is a promising future direction.
>
> We assume our primitive policies return a signal that indicates whether the execution should be terminated or not, similar to [1-3, 13]. Without access to this termination signal, the transition policy would learn from very sparse and delayed reward.
>
>
> [1] Oh et al. “Zero-shot task generalization with multi-task deep reinforcement learning”, ICML 2017
> [2] Andreas et al. “Modular multitask reinforcement learning with policy sketches”, ICML 2017
> [3] Kulkarni et al. “Hierarchical deep reinforcement learning: Integrating temporal abstraction and intrinsic motivation”, NIPS 2016
> [4] Strehl Littman “An analysis of model-based interval estimation for markov decision processes”, Journal of Computer and System Sciences (JCSS) 2008
> [5] Bellemare et al “Unifying Count-Based Exploration and Intrinsic Motivation”, NIPS  2016
> [6] Martin et al. “Count-Based Exploration in Feature Space for Reinforcement Learning”, IJCAI 2017
> [7] Schmidhuber “A possibility for implementing curiosity and boredom in model-building neural controllers”, From animals to animats: Proceedings of the first international conference on simulation of adaptive behavior, 1991
> [8] Pathak et al. “Curiosity-driven Exploration by Self-supervised Prediction”, ICML 2017
> [9] Achiam and Sastry “Surprise-Based Intrinsic Motivation for Deep Reinforcement Learning”, NIPS Workshop 2016
> [10] Stadie et al. “Incentivizing exploration in reinforcement learning with deep predictive models”, NIPS Workshop 2015
> [11] Hausman et al. “Learning an Embedding Space for Transferable Robot Skills”, ICLR 2018
> [12] Gudimella et al. “Deep reinforcement learning for dexterous manipulation with concept networks”, arXiv 2017
> [13] Le et al. “Hierarchical Imitation and Reinforcement Learning”, ICML 2018

---

### Official Review · AnonReviewer3 · 2018-11-05
**Useful  learning scheme for transitioning between options in continuous domains.**

**Rating:** 7
**Confidence:** 4

**Review:**

The paper proposes a scheme for transitioning to favorable starting states for executing given options in continuous domains. Two learning processes are carried out simultaneously: one learns a proximity function to favorable states from previous trajectories and executions of the option,  and the other learns the transition policies based on dense reward provided by the proximity function.

Both parts of the learning algorithms are pretty straightforward, but their combination turns out to be quite elegant. The experiments suggest that the scheme works,  and in particular does not get stuck in local minima.

The experiments involve fairly realistic robotic applications with complex options,  which renders credibility to the results.

Overall this is a nice contribution to the options literature. The scheme itself is quite simple and straightforward, but still useful.

One point that I would like to see elaborated is the choice of exponential ("discounted") proximity function. Wouldn't a linear function of "step" be
 more natural here? The exponent loses sensitivity as the number of steps away increases, which may lead to sparser rewards.

---

> ### Author Response · Authors · 2018-11-12
> **Response to Reviewer 3**
>
> We thank the reviewer for the feedback and address the concerns in detail below.
>
> > Reviewer 3 (R3): “... the choice of exponential (“discounted”) proximity function. Wouldn’t a linear function of “step” be more natural here?”
>
> The proximity predictor is used to reward the ending state of a transition trajectory in how close it is to the initiation set of the next primitive as well as actions that increase proximity. As R3 suggested, both linear and exponential functions are valid choices for a proximity function.
>
> We have experimentally compared the linear and exponential proximity functions. Our model is able to learn well with both functions and they perform similarly. We added the result to our website (Ablation study on Proximity functions: https://sites.google.com/view/transitions-iclr2019#h.p_qGO2W2Dk2q8G ) and will it add to the supplementary.
>
> Originally, we opted for the exponential proximity function with the intuition that the faster initial decay near the initiation set would help the policy discriminate successful states from failing states near the initiation set. Also, in our experiments, as we use 0.95 as a decaying factor, the proximity is still reasonably large (e.g., 0.35 for 20 time-steps and 0.07 for 50 time-steps).

---

### Comment · Area_Chair1 · 2018-11-20
**reviews & responses are appreciated;  remaining consideration of author responses?**

The reviews and author responses are appreciated.
If there are any further comments from the reviewers with regard to the authors responses, or changes in score,
now would be the time to put these forward.

thanks again for your insights.
-- area chair

---

> ### Comment · AnonReviewer3 · 2018-11-24
> **Score raised fro 7 to 8**
>
> The authors have responded thoroughly to the review comments. Also, looking at the simulations more closely they appear quite effective.

---

### Meta-Review · Area_Chair1 · 2018-12-14
**Well motivated problem; good solution**

**Confidence:** 5
**Recommendation:** Accept (Poster)

**Metareview:**

Strengths: The paper tackles a novel, well-motivated problem related to options & HRL.
The problem is that of learning transition policies, and the paper proposes
a novel and simple solution to that problem, using learned proximity predictors and transition
policies that can leverage those. Solid evaluations are done on simulated locomotion and
manipulation tasks. The paper is well written.

Weaknesses: Limitations were not originally discussed in any depth.
There is related work related to sub-goal generation in HRL.
AC: The physics of the 2D walker simulations looks to be unrealistic;
the character seems to move in a low-gravity environment, and can lean
forwards at extreme angles without falling. It would be good to see this explained.

There is a consensus among reviewers and AC that the paper would make an excellent ICLR contribution.
AC: I suggest a poster presentation; it could also be considered for oral presentation based
on the very positive reception by reviewers.